# Unleash Data Generation for Efficient and Effective Data-free Knowledge Distillation

## Abstract

Data-Free Knowledge Distillation (DFKD) has recently made remarkable advancements with its core principle of transferring knowledge from a teacher neural network to a student neural network without requiring access to the original data. Nonetheless, existing approaches encounter a significant challenge when attempting to generate samples from random noise inputs, which inherently lack meaningful information. Consequently, these models struggle to effectively map this noise to the ground-truth sample distribution, resulting in the production of low-quality data and imposing substantial time requirements for training the generator. In this paper, we propose a novel Noisy Layer Generation method (NAYER) which relocates the randomness source from the input to a noisy layer and utilizes the meaningful constant label-text embedding (LTE) as the input. The significance of LTE lies in its ability to contain substantial meaningful inter-class information, enabling the generation of high-quality samples with only a few training steps. The language model just is used once to query LTE, and then LTE is stored in memory for all subsequent training processes. Simultaneously, the noisy layer plays a key role in addressing the issue of diversity in sample generation by preventing the model from overemphasizing the constrained label information. By reinitializing the noisy layer in each iteration, we aim to facilitate the generation of diverse samples while still retaining the method's efficiency, thanks to the ease of learning provided by LTE. Experiments carried out on multiple datasets demonstrate that our NAYER not only outperforms the state-of-the-art methods but also achieves speeds 5 to 15 times faster than previous approaches. The code is available at `https://github.com/fw742211/nayer`.

## 1 Introduction

Knowledge distillation (KD) aims to train a student model capable of emulating the capabilities of a pre-trained teacher model. Over the past decade, KD has been explored across diverse domains, including image recognition (Qiu et al., 2022), speech recognition (Yoon et al., 2021), and natural language processing (Sanh et al., 2019). Conventional KD methods generally assume that the student model has access to all or part of the teacher's training data. However, real-world applications often impose constraints on accessing the original training data. This issue becomes particularly relevant in cases involving privacy-sensitive medical data, which may contain personal information or data considered proprietary by vendors. Consequently, in such contexts, conventional KD methods no longer suffice to address the challenges posed.

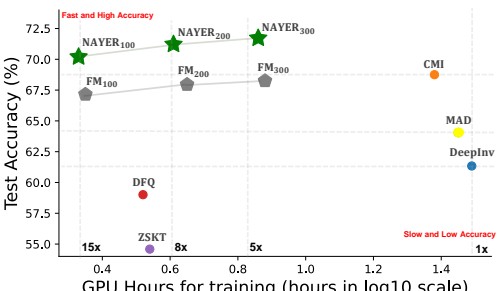

Figure 1: Accuracy of student models and GPU hours of training time on CIFAR-100 dataset. All variants of our method NAYER not only attains the highest accuracies across but also accelerates the training process by 5 to 15 times compared to DeepInv (Yin et al., 2020).

Data-Free Knowledge Distillation (DFKD) has recently seen significant advancements as an alternative method. Its core principle involves transferring knowledge from a teacher neural network ($\mathcal{T}$) to

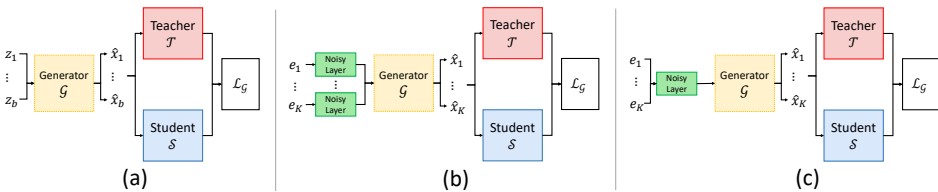

Figure 2: Data Generation Strategies: (a) The classic Data-free Generation, which optimizes random noise (z); (b) the 1-to-1 Noisy Layer Generation, which uses one noisy layer for generating one synthetic image from the embedding of label's text ($e_y$) by CLIP (Radford et al., 2021); (c) K-to-1 Noisy Layer Generation, which uses one noisy layer to generate multiple synthetic images.

a student neural network ($\mathcal{S}$) by generating synthetic data instead of accessing the original training data. The synthetic data enable adversarial training of the generator and student (Nayak et al., 2019; Micaelli & Storkey, 2019). In this setup, the student seeks to match the teacher's predictions on synthetic data, while the generator aims to create samples that maximize the discrepancy between the student's and teacher's predictions (Fig. 2a).

Due to its reliance on synthetic samples, the need for an effective and efficient data-free generation technique becomes imperative. A major limitation of current DKFD methods is that they merely generate synthetic samples from random noise, neglecting to incorporate supportive and semantic information (Binici et al., 2022a; Fang et al., 2022; Yu et al., 2023; Patel et al., 2023). This limitation in turn incurs the generation of low-quality data and excessive time requirements for training the generator, rendering them unsuitable for large-scale tasks. Notably, almost SOTA DFKD methods do not report results on large-scale ImageNet due to the significant training time involved. Even with smaller datasets such as CIFAR-100 (see Fig. 1), state-of-the-art DFKD methods such as CMI (Fang et al., 2021), MAD (Do et al., 2022), or DeepInv still demand approximately 25 to 30 hours of training while struggling to achieve high accuracy. This emphasizes the pressing need for more efficient and effective DFKD techniques.

To address mentioned problem, we introduce a simple yet effective DFKD method called **N**oisy **LAYER** Generation (NAYER). Our approach relocates the source of randomness from the input to the noisy layer and utilizes the meaningful label-text embedding (LTE) generated by a pretrained language model (LM) (Reimers & Gurevych, 2019; Radford et al., 2021) as the input. In this context, LTE plays a crucial role in accelerating the training process due to its ability to encapsulate useful interclass information. It is noteworthy that in the field of text embedding, there is a common observation that the text with similar meanings tend to exhibit closer embedding proximity to one another (Le & Mikolov, 2014). From that, the text embedding of sentence *"A class of a dog"* and *"A class of a cat"* is always closer compared to *"A class of a car"*. Consequently, by using LTE as input, our approach can proficiently generate high-quality samples that closely mimic the distributions of their respective classes with only a few training steps. Note that, to ensure a data-free setting, our method only queries the LTE from the pretrained language model once. This LTE is then stored in memory for subsequent processing, and we do not use the language model in the training process.

However, when utilizing LTE as the input, we empirically observed that existing methods suffer from a form of mode collapse. This means the generator consistently produces similar data in every iteration. A naive approach to address this is to consider the concatenation of LTE and a vector of random noise as the input for the generator. Unfortunately, it does not help in this case. We attribute this phenomenon to an overemphasis on constant label-related information. This implies that if the model has two sources of input, the first one remains unchanged and has discriminator ability, while the other changes every iteration. The model tends to focus on learning the first source and ignores the second. In DFKT, this emphasis might inadvertently overshadow the crucial random noise component necessary for generating a diverse array of samples.

Our solution addresses this issue by relocating the source of randomness from the input to the layer level by adding a noisy layer to learn the constant label information. This involves incorporating a random noise layer to function as an intermediary between the generator and LTE, which prevents the generator from relying solely on unchanging label information (Fig. 2b). The source of randomness now comes from the random reinitialization of the noisy layer for each iteration. Through this mechanism, we aim to effectively mitigate the risk of overemphasizing label information, thus enhancing the diversity of synthesized images. Furthermore, thanks to the inherent ease of learning

label-text embeddings, regardless of how it is initialized, the noisy layer can consistently generate high-quality samples in just a few steps, thereby maintaining the method's efficiency. Additionally, we propose leveraging a single noisy layer to generate multiple samples (e.g., 100 images across 100 classes of CIFAR-100) (Fig. 2c). This strategy capitalizes on the multiple gradient sources stemming from various classes, enhancing the diversity of the noisy layer's output, reducing model size and expediting the training process.

We conducted comprehensive experiments on several datasets, including ImageNet, demonstrating the superiority of our proposed techniques over state-of-the-art algorithms. Notably, NAYER not only outperforms the existing state-of-the-art approaches in terms of accuracy but also exhibits remarkable speed enhancements. Specifically, as illustrated in Fig. 1, our proposed methods achieve speeds that are 5 to even 15 times faster while also attaining higher accuracies compared to previous methods, highlighting their efficiency and effectiveness.

## 2 RELATED WORK

**Data-Free Knowledge Distillation.** DFKD methods generate synthetic images to transfer knowledge from a pre-trained teacher model to a student model. These data are used to jointly train the generator and the student in an adversarial manner (Nayak et al., 2019; Micaelli & Storkey, 2019). Under this adversarial learning scheme, the student attempts to make predictions close to the teacher's on synthetic data, while the generator tries to create samples that maximize the mismatch between the student's and the teacher's predictions. This adversarial game enables a rapid exploration of synthetic distributions useful for knowledge transfer between the teacher and the student.

**Data-Free Generation.** As the central principle of DFKD revolves around synthetic samples, the data-free generation technique plays a pivotal role. (Yin et al., 2020) proposes the image-optimized method which attempts to optimize the random noise images using teacher network batch normalization statistics. Sample-optimized methods (Fang et al., 2021; Yu et al., 2023) focus on optimizing random noise over numerous training steps to produce synthetic images in case-by-case strategy. In contrast, generator-optimized methods (Do et al., 2022; Patel et al., 2023; Binici et al., 2022a) attempt to ensure that the generator has the capacity to comprehensively encompass the entire distribution of the original data. In the other words, regardless of the input random noise, these methods aim to consistently yield high-quality samples for training the student model. This approach often prolongs the training process and may not consistently produce high-quality samples, particularly when diverse noises are employed during both the sampling and training phases. Furthermore, the main problem in existing data-free generation is the use of random noise input without any meaningful information, leading to generate the low-quality samples and prolonged training times for the generator. (Fang et al., 2022) introduced FM, a method incorporating a meta generator to accelerate the DFKD process significantly. However, this acceleration comes at the cost of a noticeable trade-off in classification accuracy. Also, several methods Wang et al. (2023); Chen et al. (2021) utilize additional unlabeled data from the wild to enhance the performance of DFKD. However, in cases involving sensitive or private data, collecting suitable unlabeled sources can be challenging, limiting their application.

**Synthetizing Samples from Label Information**. Drawing inspiration from the success of incorporating label information in adversarial frameworks like Conditional GAN (Mirza & Osindero, 2014), several methods in DFKD have adopted strategies to generate images guided by labels. In these approaches, a common practice involves fusing random noise ($z$) with a learnable embedding ($e_y$) of the one-hot label vector, which is used as input for the model (Luo et al., 2020; Yoo et al., 2019; Do et al., 2022). This combination enhances control over the resulting class-specific synthetic images. However, despite the potential of label information, its application has yielded only minor improvements. This can be attributed to two key factors. Firstly, the one-hot vector introduces sparse information that merely distinguishes labels uniformly, failing to capture the nuanced relationships between different classes. Consequently, the model struggles to generate images that align closely with ground-truth distributions. Secondly, there exists a challenge in balancing the generated images' quality and diversity when incorporating label information. This can inadvertently lead to an overemphasis on label-related details, potentially overshadowing the crucial contribution of random noise, which is necessary for generating a diverse range of samples.

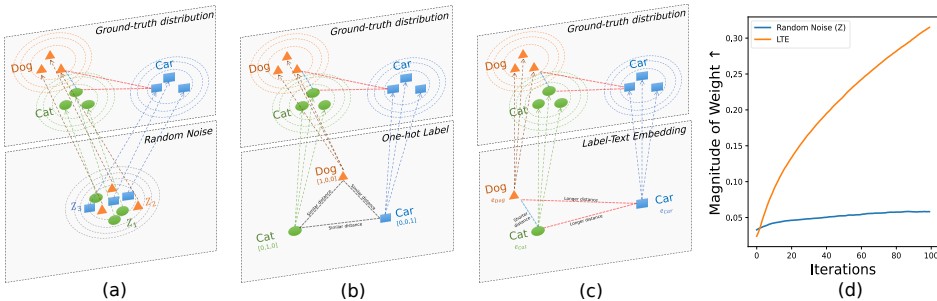

Figure 3: (a) Random noise for data generation. (b) One-hot labels only uniformly distinguish labels, lacking inter-class relationships. In contrast, (c) LTE captures inter-class connections, bringing similar classes closer in the embedding space. This proximity enhances the similarity between the input and ground-truth sample distributions, thereby allowing the model to more easily mimic the ground-truth distribution and accelerating the learning process. (d) The averaging magnitude of weight used to learn LTE is much larger than those for random noise, highlighting the model's negative focus on label information while ignoring random noise.

## 3  PROPOSED METHOD

### 3.1  PROBLEM FORMULATION

Consider a training dataset $D = \{(\boldsymbol{x}_i, \boldsymbol{y}_i)\}_{i=1}^{m}$ with $\boldsymbol{x}_i \in \mathbb{R}^{c \times h \times w}$ and $\boldsymbol{y}_i \in \{1, 2, \cdots, K\}$, where the pair $(\boldsymbol{x}_i, \boldsymbol{y}_i)$ represents a training sample and its corresponding label, respectively. Let $\mathcal{T} = \mathcal{T}_{\theta_{\mathcal{T}}}$ be a pre-trained teacher network on $D$. The objective of DFKD is to train a student network $\mathcal{S} = \mathcal{S}_{\theta_{\mathcal{S}}}$ to emulate $\mathcal{T}$'s performance, all without needing access to the original dataset $D$.

To achieve this, we employ the lightweight generator $\mathcal{G}_{\theta_{\mathcal{G}}}$ to generate synthetic images and subsequently use them to train a student network $\mathcal{S}$. Specifically, in contrast to existing DFKD methods Yu et al. (2023); Do et al. (2022); Patel et al. (2023); Yin et al. (2020); Fang et al. (2021; 2022), our approach utilizes a meaningful constant label-text embedding (LTE) as the input for $\mathcal{G}$ instead of random noise. Due to LTE's capability to encapsulate valuable interclass information, this accelerates the generation process, expediting the training time (Section 3.2). Following that, we propose the use of a layer-level random source (Noisy Layer) to better adapt with LTE for generating diverse synthetics (Section 3.3). Finally, the synthetic images are employed for the joint training of the generator and student in an adversarial manner to enhance knowledge transfer (Section 3.4).

### 3.2  LABEL-TEXT EMBEDDING AS GENERATOR'S INPUT

The main limitation of existing DFKD methods is synthetize data from random noise, which have no supportive and semantic information. Therefore, they usually generate very low-quality data Fang et al. (2022) or require a excessive training time for high quality image generation Do et al. (2022); Patel et al. (2023); Yu et al. (2023); Fang et al. (2021); Yin et al. (2020). There are also several methods use to one-hot vector of classes as the additional input to resemble the conditional generator, however its application has yielded only minor improvements. The main reason is the one-hot vector (OH) introduces sparse information which make a generator hard to learn about it. Furthermore, OH merely distinguishes labels uniformly, failing to capture the nuanced relationships between different classes.

To address this problem, we are the first to propose the use of label-text embeddings for DFKD by employing them as an input for the generator. LTE, as a dense vector with richer information, facilitates an easier learning process for the model. Additionally, LTE capitalizes on the tendency for text with similar meanings to exhibit proximity in their embeddings Le & Mikolov (2014). Figure 3a-c visually represents the LTE, highlighting their superior capacity to depict the relationship between the 'Dog' and 'Cat' classes. This is evident in their closer proximity (shorter distance) when compared to the 'Car' class. This characteristic of LTE contributes to making the input distribution (representing labels) and ground-truth distribution (representing actual data) more similar. As a result, it facilitates the model's mapping between these two distributions, accelerating the learning process and generating high-quality images.

**Prompt Engineering.** Given the list of all classes $\boldsymbol{y} = [\boldsymbol{y}_1, \cdots, \boldsymbol{y}_K]$, their label text $Y_{\boldsymbol{y}} = [Y_{\boldsymbol{y}_1}, \cdots, Y_{\boldsymbol{y}_K}]$ is generated by using a manually designed prompt template such as `"a photo of a {class_name}"`. Then, the label-text prompt is then embedded using a pre-trained text encoder $\mathcal{C}$ as follows:

$$\boldsymbol{e}_{\boldsymbol{y}} = \mathcal{C}(Y_{\boldsymbol{y}}) . \tag{1}$$

**LTE Pool.** Importantly, the embedding $\boldsymbol{e}_{\boldsymbol{y}}$ is generated once and then stored in the LTE pool $\mathcal{P}$, remaining fixed throughout the entire training process. The text encoder $\mathcal{C}$ is not utilized during the training process. In training phases, with a batch of pseudo-labels $\hat{\boldsymbol{y}}$, we retrieve their corresponding LTEs from $\boldsymbol{e}_{\hat{\boldsymbol{y}}} \sim \mathcal{P}$ and employ these LTEs as inputs for the generator. This eliminates the reliance on random noise for synthetic image generation.

$$\hat{\boldsymbol{x}} = \mathcal{G}(\boldsymbol{e}_{\hat{\boldsymbol{y}}}) . \tag{2}$$

We conducted an ablation study to analyze the impact of different prompt engineering template and language model (LM) for generating LTEs in Section 4.4.

Thanks to the informative content embedded in LTE, our approach can efficiently produce high-quality samples with minimal computational steps. We have also conducted an empirical study to substantiate this claim, as illustrated in Table 3. The results of this study highlight that LTE significantly accelerates convergence in terms of Cross-Entropy (CE) Loss and yields higher-quality images (as measured by the Inception Score or IS score) compared to random noise and one-hot vectors. This acceleration empowers our method to achieve convergence with a considerably smaller training steps for generator (30 steps for CIFAR10 and 40 steps for CIFAR100), compared to the 2,000 steps required by DeepInv or the 500 steps of CMI, all while maintaining superior accuracy (as detailed in Table 1).

### 3.3 GENERATING DIVERSE SAMPLES WITH NOISY LAYER

While leveraging label information provides advantages for data generation, the synthetic images are less diverse due to the absence of a random source. Two common solutions involve concatenating random noise $\boldsymbol{z}$ and $\boldsymbol{e}_{\boldsymbol{y}}$ or using their sum as the generator input. However, both approaches have limitations. Concatenation raises the risk of overemphasizing label-related information, as evidenced by significantly larger weight magnitudes for learning LTE compared to random noise (Figure 3d), which can be seen as the significance of these weights Frankle & Carbin (2018). Using the sum of $\boldsymbol{v} = \boldsymbol{e}_{\hat{\boldsymbol{y}}} + \beta \boldsymbol{z}$ faces challenges: a low $\beta$ results in an insufficient random source for diverse sampling, and a high $\beta$ may overshadow LTE features, leading to a reliance on random noise $\boldsymbol{z}$. This challenge is also observed in some existing methods Luo et al. (2020); Do et al. (2022), where the application of the sum of noise and label information provides minimal improvement compared to an unconditional generator. To effectively introduce randomness to LTE, we propose the concept of a layer-level random source with the Noisy Layer. The source of randomness now stems from the random reinitialization of the NL during each iteration. With each different initialization, the NL learns LTE in a distinct way, successfully mitigating the risk of a negative bias towards LTE. Unlike existing sources of randomness, the design of NL provides a larger random parameter to enhance the diversity of the synthesized images. Furthermore, due to the straightforward training of LTE, regardless of its initialization, the joint training of the noisy layer and the generator consistently yields high-quality samples within a few iterations, thus preserving the method's efficiency.

**Noisy Layer Architecture.** We design the NL $\mathcal{Z}_{\theta_{\mathcal{Z}}}$ as a combination of a `BatchNorm` layer and a single `Linear` layer. The input size of the `Linear` layer matches the embedding size of the text encoder ($e$), and the output size corresponds to the noise dimension ($r$). Typically, this output size is set to 1,000, following to Patel et al. (2023); Do et al. (2022); Yu et al. (2023). The simplicity of the single `Linear` layer is crucial for expediting the generation process. It converges rapidly without requiring an excessive number of steps, yet its size remains sufficiently large to provide an sufficient random source for the generator. Additionally, a `BatchNorm` module plays a role in increasing the distance between LTEs (from averaging 0.015 to 0.45 using L2 distance), helping the model discriminate these LTEs easier and thereby speeding up the training process. Furthermore, with a different batch of $\hat{\boldsymbol{y}}$, the output of `BatchNorm` can vary, introducing a slight additional randomness for the generator. The ablation study analyzing the impact of different architectures of NL can be found in the Supplemental Material.

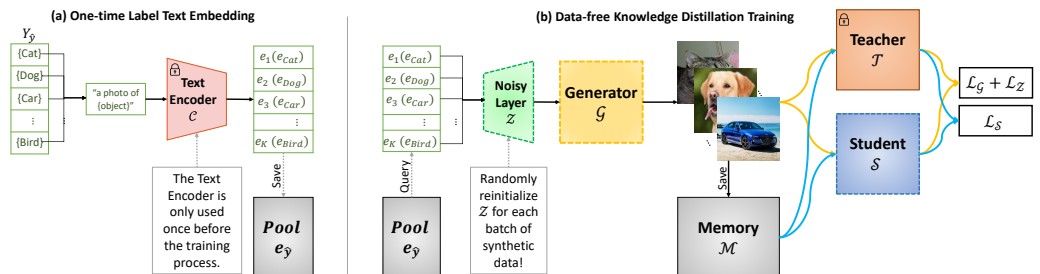

Figure 4: General Architecture of Noisy Layer Generation for Data-free Knowledge Distillation: NAYER initially employs the text encoder to generate the LTEs, which are then stored in the memory pool for model training. In each training batch, the LTEs serve as input for the noisy layer $\mathcal{Z}$ and generator $\mathcal{G}$ to produce synthetic images. Finally, these images are used for the joint training of the generator, noisy layer, and student network using Eq. 5 and Eq. 6.

---

**Algorithm 1: NAYER**

**Input:** pre-trained teacher $\mathcal{T}_{\theta_{\mathcal{T}}}$, student $\mathcal{S}_{\theta_{\mathcal{S}}}$, generator $\mathcal{G}_{\theta_{\mathcal{G}}}$, text encoder $\mathcal{C}_{\theta_{\mathcal{C}}}$, list of labels $\boldsymbol{y}$ and list of text of these labels $Y_{\boldsymbol{y}}$;

**Output:** An optimized student $\mathcal{S}_{\theta_{\mathcal{S}}}$

1 Initializing $\mathcal{P} = \{\}, \mathcal{M} = \{\}$;
2 Store all embeddings $\boldsymbol{e_y} = \mathcal{C}(Y_{\boldsymbol{y}})$ into $\mathcal{P}$;
3 **for** $\mathcal{E}$ *epochs* **do**
4     **for** $I$ *iterations* **do**
5         Randomly reinitializing noisy layers $\mathcal{Z}_{\theta_{\mathcal{Z}}}$ and pseudo label $\hat{\boldsymbol{y}}$ for each iteration;
6         Query $\boldsymbol{e_{\hat{y}}} \sim \mathcal{P}$;
7         **for** $g$ *steps* **do**
8             $\hat{\boldsymbol{x}} \leftarrow \mathcal{G}(\mathcal{Z}(\boldsymbol{e_{\hat{y}}}))$;
9             $\mathcal{L}_{\mathcal{Z}} \leftarrow \alpha_{cls}\mathcal{L}_{\text{CE}}(\mathcal{T}(\hat{\boldsymbol{x}}), \hat{\boldsymbol{y}}) - \alpha_{adv}\mathcal{L}_{\text{KL}}(\mathcal{T}(\hat{\boldsymbol{x}}), \mathcal{S}(\hat{\boldsymbol{x}})) + \alpha_{bn}\mathcal{L}_{\text{BN}}(\mathcal{T}(\hat{\boldsymbol{x}}))$;
10             Update $\theta_{\mathcal{G}}, \theta_{\mathcal{Z}}$ by minimizing $\mathcal{L}_{\mathcal{Z}}$;
11         $\mathcal{M} \leftarrow \mathcal{M} \cup \hat{\boldsymbol{x}}$;
12     **for** $S$ *iterations* **do**
13         $\hat{\boldsymbol{x}} \sim \mathcal{M}$;
14         Update $\theta_{\mathcal{S}}$ by minimizing $\mathcal{L}_{\mathcal{S}} \leftarrow \mathcal{L}_{\text{KL}}(\mathcal{T}(\hat{\boldsymbol{x}}), \mathcal{S}(\hat{\boldsymbol{x}}))$;

---

Given LTEs $\boldsymbol{e_{\hat{y}}}$, we feed these $\boldsymbol{e_{\hat{y}}}$ into the noisy layer $\mathcal{Z}$. Then, the output of the noisy layer is fed into the generator $\mathcal{G}$ to produce the batch of synthetic images $\hat{\boldsymbol{x}}$:

$$\mathcal{Z}(\boldsymbol{e_{\hat{y}}}) = \texttt{Linear}(\texttt{BatchNorm}(\boldsymbol{e_{\hat{y}}})) . \tag{3}$$

$$\hat{\boldsymbol{x}} = \mathcal{G}(\mathcal{Z}(\boldsymbol{e_{\hat{y}}})) . \tag{4}$$

**K-to-1 Noisy Layer.** In the existing approach, a separate random source is created for each instance, as similar inputs generate similar samples. In contrast, we propose employing a single noisy layer to learn from all available classes (K-to-1) by inputting $\boldsymbol{e_{\hat{y}}}$ with $\hat{\boldsymbol{y}} = 1, \ldots, K$ to a single noisy layer $\mathcal{Z}$. This design enables the noisy layer to generate multiple samples simultaneously, such as a maximum of 100 for CIFAR100 or 10 for CIFAR10, thus reducing a parameter size and efficiently expediting training. The underlying idea revolves around the fact that each class has distinct LTEs. Thus, by supplying different inputs of $\boldsymbol{e_{\hat{y}}}$ from K classes, the noisy layer can still generate diverse images. Furthermore, we also empirically observe that using a single noisy layer to synthesize a batch of images (K-to-1) enriches generator diversity, ensuring both fast convergence and high-quality sample generation. This enhancement can be attributed to the use of multiple gradient sources from diverse classes, which can further enriches the diversity of the noisy layer's output.

## 3.4 GENERATOR AND STUDENT UPDATING

To make it easier to follow, we provide the architecture of NAYER in Figure 4 and the detailed pseudocode in Algorithm 1, wherein NAYER initially embeds all label text using a text encoder. Subsequently, our method undergoes training for $\mathcal{E}$ epochs. Within each training epoch, NAYER consists of two distinct phases. The first phase involves training the generator. In each iteration $I$, as described in Algorithm 1, the noisy layer $\mathcal{Z}$ is reinitialized (line 5) before being utilized to learn

the LTE. The generator and the noisy layer are then trained through $g$ steps using Eq. (5) to optimize their performance (line 10).

$$\min_{\theta_{\mathcal{G}}, \theta_{\mathcal{Z}}} \mathcal{L}_{\mathcal{Z}} \triangleq \mathbb{E}_{\hat{\boldsymbol{x}} \sim \mathcal{G}(\mathcal{Z}(\boldsymbol{e}_{\hat{\boldsymbol{y}}}))} \Big[ \alpha_{cls} \mathcal{L}_{\text{CE}}(\mathcal{T}(\hat{\boldsymbol{x}}), \hat{\boldsymbol{y}}) - \alpha_{adv} \mathcal{L}_{\text{KL}}(\mathcal{T}(\hat{\boldsymbol{x}}), \mathcal{S}(\hat{\boldsymbol{x}})) + \alpha_{bn} \mathcal{L}_{\text{BN}}(\mathcal{T}(\hat{\boldsymbol{x}})) \Big] . \quad (5)$$

Within this context, $\mathcal{L}_{\text{CE}}$ represents the Cross-Entropy loss term, serving the purpose of training the student on images residing within the high-confidence region of the teacher's knowledge. Conversely, the negative $\mathcal{L}_{\text{KL}}$ term facilitates the exploration of synthetic distributions, boosting effective knowledge transfer between the teacher and the student. In other words, the student network takes on a role as a discriminator in GANs, ensuring the generator is geared towards producing images that the teacher has mastered, yet the student network has not previously learned. This approach facilitates the focused development of the student's understanding in areas where it lags behind the teacher, enhancing the overall knowledge transfer process. We also use batch norm regularization ($\mathcal{L}_{\text{BN}}$) Yin et al. (2020); Fang et al. (2022), a commonly used loss in DFKD, to constrain the mean and variance of the feature at the `BatchNorm` layer to be consistent with the running-mean and running-variance of the same layer.

The second phase involves training the student networks. During this phase, all the generated samples are stored in the memory module $\mathcal{M}$ to mitigate the risk of forgetting (line 10), following a similar approach as outlined in Fang et al. (2022). Ultimately, the student model is trained by Eq. (6) over $S$ iterations, utilizing the samples from $\mathcal{M}$ (lines 13 and 14).

$$\min_{\theta_{\mathcal{S}}} \mathcal{L}_{\mathcal{S}} \triangleq \mathbb{E}_{\hat{\boldsymbol{x}} \sim \mathcal{M}} \Big[ \mathcal{L}_{\text{KL}}(\mathcal{T}_{\theta_{\mathcal{T}}}(\hat{\boldsymbol{x}}), \mathcal{S}_{\theta_{\mathcal{S}}}(\hat{\boldsymbol{x}})) \Big] . \quad (6)$$

## 4 EXPERIMENTS

### 4.1 EXPERIMENTAL SETTINGS

We conducted a comprehensive evaluation of our method across various backbone networks, namely ResNet (He et al., 2016), VGG (Simonyan & Zisserman, 2014), and WideResNet (WRN)(Zagoruyko & Komodakis, 2016), spanning three distinct classification datasets: CIFAR10, CIFAR100 (Krizhevsky et al., 2009), and Tiny-ImageNet (Le & Yang, 2015). The datasets feature varying scales and complexities, offering a well-rounded assessment of our method's capabilities. In detail, CIFAR10 and CIFAR100 encompass a total of 60,000 images, partitioned into 50,000 for training and 10,000 for testing. CIFAR10 comprises 10 categories, while CIFAR100 boasts 100 categories. The images within both datasets are characterized by a resolution of 32×32 pixels. On the other hand, Tiny-ImageNet comprises 100,000 training images and 10,000 validation images, with a higher resolution of $64 \times 64$ pixels. This dataset encompasses a diverse array of 200 image categories, contributing to the breadth and comprehensiveness of our evaluation.

### 4.2 RESULTS AND ANALYSIS

**Comparison with SOTA DFKD Methods.** Table 1 displays the results of DFKD achieved by our methods and several state-of-the-art (SOTA) approaches. In general, previous methods exhibit limitations when generating images from random noise, impacting both training time and image diversity. By using LTE as the input and relocating the source of randomness from the input to the layer level, our approach provides highly diverse training images and faster running time. Notably, with 300 epochs, our method achieves SOTA performance in all comparison cases, except for the Resnet32/Resnet18 case in CIFAR10. However, it is essential to note that our method was designed in a straightforward manner, without incorporating innovative techniques found in current SOTA approaches, such as activation region constraints and feature exchange in SpaceshipNet (Yu et al., 2023), knowledge acquisition and retention meta-learning in KAKR (Patel et al., 2023), and momentum distillation in MAD (Do et al., 2022).

**Additional Experiments at Higher Resolution.** To assess the effectiveness of NAYER, we conducted further evaluations on the more challenging ImageNet dataset. ImageNet comprises 1.3 million training images with resolutions of 224×224 pixels, spanning 1,000 categories. ImageNet's complexity surpasses that of CIFAR, making it a significantly more time-consuming task for data-free training. As displayed in Table 1, almost all DFKD methods refrain from reporting results

Table 1: The accuracies of compared methods. The best-performing method is highlighted in bold, and the runner-up is underlined. Additionally, we use superscripts to indicate the sources of these results: [a] for Fang et al. (2022), [b] for Patel et al. (2023), [c] for Do et al. (2022), [d] for Yu et al. (2023), and [e] for our experiments. In this table, 'R' represents Resnet, 'W' corresponds to WideResnet, and 'V' stands for VGG.

| Method | CIFAR10 | | | | | CIFAR100 | | | | | TinyImageNet | ImageNet |
| | R34 R18 | W402 W162 | W402 W161 | W402 W401 | V11 R18 | R34 R18 | W402 W162 | W402 W161 | W402 W401 | V11 R18 | R34 R18 | R50 R50 |
| --- | --- | --- | --- | --- | --- | --- | --- | --- | --- | --- | --- | --- |
| Teacher | 95.70 | 94.87 | 94.87 | 94.87 | 92.25 | 77.94 | 77.83 | 75.83 | 75.83 | 71.32 | 66.44 | 75.45 |
| Student | 95.20 | 93.95 | 91.12 | 93.94 | 95.20 | 77.10 | 73.56 | 65.31 | 72.19 | 77.10 | 64.87 | 75.45 |
| DeepInv[a] (Yin et al., 2020) | 93.26 | 89.72 | 83.04 | 86.85 | 90.36 | 61.32 | 61.34 | 53.77 | 68.58 | 54.13 | - | 68.00 |
| DFQ[a] (Choi et al., 2020) | 94.61 | 92.01 | 86.14 | 91.69 | 90.84 | 77.01 | 64.79 | 51.27 | 54.43 | 66.21 | - | - |
| ZSKT[a] (Micaelli & Storkey, 2019) | 93.32 | 89.66 | 83.74 | 86.07 | 89.46 | 67.74 | 54.59 | 36.60 | 53.60 | 54.31 | - | - |
| CMI[a] (Fang et al., 2021) | 94.84 | 92.52 | 90.01 | 92.78 | 91.13 | 77.04 | 68.75 | 57.91 | 68.88 | 70.56 | 64.01 | - |
| PREKD[b] (Binici et al., 2022a) | 93.41 | - | - | - | - | 76.93 | - | - | - | - | 49.94 | - |
| MBDFKD[b] (Binici et al., 2022b) | 93.03 | - | - | - | - | 76.14 | - | - | - | - | 47.96 | - |
| FM[a] (Fang et al., 2022) | 94.05 | 92.45 | 89.29 | 92.51 | 90.53 | 74.34 | 65.12 | 54.02 | 63.91 | 67.44 | - | 57.37[e] |
| MAD[c] (Do et al., 2022) | 94.90 | 92.64 | - | - | - | 77.31 | 64.05 | - | - | - | 62.32 | - |
| KAKR_MB[b] (Patel et al., 2023) | 93.73 | - | - | - | - | 77.11 | - | - | - | - | 47.96 | - |
| KAKR_GR[b] (Patel et al., 2023) | 94.02 | - | - | - | - | 77.21 | - | - | - | - | 49.88 | - |
| SpaceshipNet[d] (Yu et al., 2023) | **95.39** | 93.25 | 90.38 | 93.56 | _92.27_ | _77.41_ | 69.95 | 58.06 | 68.78 | 71.41 | _64.04_ | - |
| **NAYER** ($\mathcal{E} = 100$) | 94.03 | 93.48 | 91.12 | 93.57 | 91.34 | 76.29 | 70.20 | 59.26 | 69.89 | 71.10 | 61.71 | - |
| **NAYER** ($\mathcal{E} = 200$) | 94.89 | _93.84_ | 91.60 | _94.03_ | 91.93 | 77.07 | _71.22_ | _61.90_ | _70.68_ | _71.53_ | 63.12 | - |
| **NAYER** ($\mathcal{E} = 300$) | _95.21_ | **94.07** | **91.94** | **94.15** | **92.37** | **77.54** | **71.72** | **62.23** | **71.80** | **71.75** | **64.17** | **68.92** |

Table 2: Comparing training times in hours using a single NVIDIA A100 for DFKD methods on CIFAR-10 and CIFAR-100 with the teacher/student models WRN40-2/WRN16-2. FM ($\mathcal{E} = 100, 200,$ and $300$) corresponds to the settings of three variants of our methods. We were unable to replicate the training times of KAKR and SpaceshipNet as they did not provide access to their source code.

| | DeepInv | CMI | DFQ | ZSKT | MAD | FM $\mathcal{E} = 100$ | FM $\mathcal{E} = 200$ | FM $\mathcal{E} = 300$ | **NAYER** $\mathcal{E} = 100$ | **NAYER** $\mathcal{E} = 200$ | **NAYER** $\mathcal{E} = 300$ |
| --- | --- | --- | --- | --- | --- | --- | --- | --- | --- | --- | --- |
| **CIFAR10** | 89.72 (31.23h) | 92.52 (24.01h) | 92.01 (3.31h) | 89.66 (3.44h) | 92.64 (13.13h) | 91.63 (2.18h) | 92.05 (3.98h) | 92.31 (7.02h) | **93.48** **(2.05h)** | 93.84 (3.85h) | 94.07 (6.78h) |
| **CIFAR100** | 61.34 (31.23h) | 68.75 (24.01h) | 64.79 (3.31h) | 54.59 (3.44h) | 64.05 (26.45h) | 67.15 (2.23h) | 67.75 (4.42h) | 68.25 (7.56h) | **70.20** **(2.15h)** | 71.22 (4.03h) | 71.72 (7.22h) |
| **Avergaing Speed Up** | $1\times$ | $1.3\times$ | $9.73\times$ | $9.08\times$ | $1.78\times$ | $14.17\times$ | $7.46\times$ | $4.29\times$ | **14.88$\times$** | $7.93\times$ | $4.47\times$ |

on ImageNet due to their prolonged training times. Therefore, our comparison is primarily against DeepInv (Yin et al., 2020), and for the sake of a fair comparison, we re-conducted the experiments of FM (Fang et al., 2022) to align with our settings. The results clearly demonstrate that NAYER outperforms other methods in terms of accuracy, underscoring its efficacy on a large-scale dataset.

**Training Time Comparison.** As shown in Table 2, the NAYER model trained for 100 epochs (i.e., NAYER($\mathcal{E} = 100$)) achieves an average speedup of $15\times$ compared to DeepInv, while also delivering higher accuracies. This substantial speedup is attributed to the significantly fewer steps required for generating samples (30 for CIFAR-10 and 40 for CIFAR-100) compared to DeepInv's 2000 steps. As a result, DeepInv takes over 30 hours to complete training on CIFAR-10/CIFAR-100, whereas our method only requires approximately 2 hours. These results demonstrate that our method not only achieves high accuracy but also significantly accelerates the model training process.

## 4.3 ABLATION STUDY

**Effectiveness of Label-Text Embedding.** We illustrate the impact of using LTE in comparison with random noise (Z) and one-hot vector (OH) as the inputs for the generator. As depicted in first three column in Table 3, LTE demonstrates significantly accelerated averaging convergence in terms of CE Loss. This phenomenon can be attributed to the principle that mapping between two distributions is simplified when they share greater similarity. However, the diversity metric for inputting label information (both LTE and OH) is notably lower than that of random noise. This outcome underscores the adverse effects of the generator overly focusing on constant LTEs.

Table 3: Comparison with different types of input and random sources involves accuracy in CIFAR100 with W402/W162 pair. All compared method is trained with $\mathcal{E} = 100$), diversity metric and averaging convergence time, which is the average number of epochs the generator needs to synthesize data with Cross-Entropy (CE) Loss $< 0.1$. Each method undergoes 30 generation steps and runs for 100 epochs. "-" denotes that a model cannot provide any data with CE Loss $< 0.1$.

| | OH | Z | LTE | cat | sum(0.1) | sum(0.5) | sum(1) | NAYER(woRI) | NAYER(1to1) | **NAYER(Kto1)** |
| --- | --- | --- | --- | --- | --- | --- | --- | --- | --- | --- |
| Avg. Convergence Time | 28.23 | - | 8.72 | 10.47 | 10.17 | 25.12 | - | 8.68 | 9.53 | **9.82** |
| Diversity Score | 0.013 | 0.137 | 0.016 | 0.0132 | 0.021 | 0.036 | 0.127 | 0.016 | 0.138 | **0.139** |
| Accuracy | 12.35 | 90.14 | 13.52 | 13.29 | 18.92 | 85.72 | 90.15 | 14.82 | 93.42 | **93.48** |

**Effectiveness of Noisy Layer.** We analyze the impact of multiple randomness source strategies, including our NAYER (1-to-1), NAYER (K-to-1), NAYER without reinitiation (WoRI), the concatenation of LTE and random noise Z (cat), and the sum of them (sum($\beta$)): $v = e_y + \beta Z$. Table 3 demonstrates that: 1) the sum of LTE and noise have a lower convergence time but higher accuracy and diversity if $\beta$ is high, making them similar to only using random noise $Z$. In contrast, if $\beta$ is low, the convergence time is faster but accuracy and diversity are lower, similar to only using LTE. 1) Using NAYER boosts the generator's diversity while maintaining rapid convergence and high-quality sampling. 2) Using a single noisy layer to synthesize a batch of images (K-to-1) results in faster convergence and a higher diversity score when compared to using one noisy layer for each individual image (1-to-1). 3) With reinitiation, the NAYER provides almost similar results to only using LTE, thereby highlighting the effectiveness of reinitiation strategies.

## 4.4 FURTHER ANALYSIS

**Comparison with Different Text Encoder** We analyze the accuracies of our NAYER (Noisy Label Generator) model across three distinct text encoders: Doc2Vec (Le & Mikolov, 2014), SBERT (Reimers & Gurevych, 2019), and CLIP (Radford et al., 2021). Table 4 shows that CLIP has the best results due to its multimodal nature, but the difference is minor (0.09%). This result also demonstrates that NAYER can work effectively with any pretrained language model.

Table 4: The accuracies of our NAYER with three different text encoders.

| | CIFAR-10 | | | | CIFAR-100 | | | |
|---|---|---|---|---|---|---|---|---|
| Text Encoder | SOTA | Doc2Vec | SBERT | CLIP | SOTA | Doc2Vec | SBERT | CLIP |
| Accuracy | 93.25 | 93.98 | 93.94 | **94.07** | 69.95 | 71.58 | 71.63 | **71.72** |

**Comparison with Different Prompting Engineering Strategies.** We analyze the impact of different prompting engineering techniques to generate the label text. Given the label $\hat{y}$ and the index of label $I_{\hat{y}}$, we propose three different ways to prompt the label text $Y_{\hat{y}}$, including P1: `"A class of `$\hat{y}$`."`; P3: `"A class of `$\hat{y}$`"`; P3: `"A photo of class `$I_{\hat{y}}$`."`. Table 5 demonstrates that: 1) While P2 has the best accuracy, the difference is not significant; 2) By using only the label index instead of the label name, the performance of P3 remains far better than the best baseline (93.72% and 71.17% compared to 93.25% and 69.95% for SpaceshipNet). From this, we can infer that using the label index is possible in the datasets with less meaningful labels, further showing the effectiveness of our methods in real-world applications.

Table 5: The accuracies of our NAYER with three different prompt approaches.

| | CIFAR-10 | | | | CIFAR-100 | | | |
|---|---|---|---|---|---|---|---|---|
| Text Encoder | SOTA | P1 | P2 | P3 | SOTA | P1 | P2 | P3 |
| Accuracy | 93.25 | 93.96 | **94.07** | 93.72 | 69.95 | 71.68 | **71.72** | 71.17 |

## 5 CONCLUSION

In this paper, we propose a novel Noisy Layer Generation method (NAYER) which utilizes the meaningful label-text embedding (LTE) as the input and relocates the randomness source from the input to the noisy layer. The significance of LTE lies in its ability to contain substantial meaningful information, enabling the fast generating images in only few steps. On the other hand, the use of noisy layer can help the model address the overfocus problem in using constant input information and increase significantly the diversity. Our extensive experiments on different datasets and tasks prove NAYER's superiority over other state-of-the-art methods in data-free knowledge distillation.

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

# A  TRAINING DETAILS

## A.1  TEACHER MODEL TRAINING DETAILS

In this work, we employed pretrained ResNet-34 and WideResnet-40-2 teacher models from (Fang et al., 2022) for CIFAR-10 and CIFAR-100. For Tiny ImageNet, we trained ResNet-34 from scratch using PyTorch, and for ImageNet, we utilized the pretrained ResNet-50 from PyTorch. Teacher models were trained with SGD optimizer, initial learning rate of 0.1, momentum of 0.9, and weight decay of 5e-4, using a batch size of 128 for 200 epochs. Learning rate decay followed a cosine annealing schedule.

Table 6: Generator Network ($\mathcal{G}$) Architecture for CIFAR10, CIFAR100 and TinyImageNet.

| Output | Size Layers |
|---|---|
| 1000 | Input |
| $128 \times h/4 \times w/4$ | Linear, BatchNorm1D, Reshape |
| $128 \times h/4 \times w/4$ | SpectralNorm (Conv ($3 \times 3$)), BatchNorm2D, LeakyReLU |
| $128 \times h/2 \times w/2$ | UpSample (2×) |
| $64 \times h/2 \times w/2$ | SpectralNorm (Conv ($3 \times 3$)), BatchNorm2D, LeakyReLU |
| $64 \times h \times w$ | UpSample (2×) |
| $3 \times h \times w$ | SpectralNorm (Conv ($3 \times 3$)), Sigmoid, BatchNorm2D |

Table 7: Generator Network ($\mathcal{G}$) Architecture for ImageNet.

| Output | Size Layers |
|---|---|
| 1000 | Input |
| $128 \times h/16 \times w/16$ | Linear, BatchNorm1D, Reshape |
| $128 \times h/16 \times w/16$ | SpectralNorm (Conv ($3 \times 3$)), BatchNorm2D, LeakyReLU |
| $128 \times h/8 \times w/8$ | UpSample (2×) |
| $128 \times h/8 \times w/8$ | SpectralNorm (Conv ($3 \times 3$)), BatchNorm2D, LeakyReLU |
| $128 \times h/4 \times w/4$ | UpSample (2×) |
| $64 \times h/4 \times w/4$ | SpectralNorm (Conv ($3 \times 3$)), BatchNorm2D, LeakyReLU |
| $64 \times h/2 \times w/2$ | UpSample (2×) |
| $64 \times h/2 \times w/2$ | SpectralNorm (Conv ($3 \times 3$)), BatchNorm2D, LeakyReLU |
| $64 \times h \times w$ | UpSample (2×) |
| $3 \times h \times w$ | SpectralNorm (Conv ($3 \times 3$)), Sigmoid, BatchNorm2D |

Table 8: The hyperparameters for NAYER applied to four different datasets are detailed below. Specifically, $\alpha_{cls}$, $\alpha_{bn}$, and $\alpha_{adv}$ are the hyperparameters associated with Eq. (**??**), and their values are consistent with the settings defined in (Fang et al., 2022). The variables $I$ and $S$ denote the number of iterations for generating and training the student, respectively, while $g$ represents the training steps to optimize the generator $\mathcal{G}_{\theta_\mathcal{G}}$ and the noisy layers $\mathcal{Z}$.

| | batch size (student) | batch size (generator) | $\alpha_{cls}$ | $\alpha_{bn}$ | $\alpha_{adv}$ | $I$ | $g$ | $S$ |
|---|---|---|---|---|---|---|---|---|
| CIFAR10 | 512 | 400 | 0.5 | 10 | 1.33 | 2 | 30 | 400 |
| CIFAR100 | 512 | 400 | 0.5 | 10 | 1.33 | 2 | 40 | 400 |
| TinyImageNet | 256 | 200 | 0.5 | 10 | 1.33 | 4 | 60 | 1000 |
| ImageNet | 128 | 50 | 0.1 | 0.1 | 0.1 | 20 | 100 | 2000 |

## A.2  STUDENT MODEL TRAINING DETAILS

To ensure fair comparisons, we adopt the generator architecture outlined in (Fang et al., 2022) for all experiments. Specifically, the generator architecture for CIFAR10, CIFAR100, and TinyImageNet is elaborated upon in Table 6, while the generator architecture for ImageNet is provided in Table 7. Across all experiments, we maintain a consistent approach for training the student model, employing a batch size of 512. We utilize the SGD optimizer with a momentum of 0.9 and a variable learning rate, following a cosine annealing schedule that starts at 0.1 and ends at 0, to optimize the student parameters ($\theta_\mathcal{S}$). Additionally, we employ the Adam optimizer with a learning rate of 4e-3 for optimizing the generator.We present the results in three distinct variants, each corresponding to a different value of $\mathcal{E}$: 100, 200, and 300, all incorporating a configuration of 20 warm-up epochs, in line with the settings defined in (Fang et al., 2022). Further details regarding the parameters can be found in Table 8.

# B    EXTENDED RESULTS

## B.1    ADDITIONAL EXPERIMENTS IN DATA-FREE QUANTIZATION.

To demonstrate the use of our data-free generation method in other data-free tasks, we further conduct experiments in Data-free Quantization. We conducted a comparative analysis against ZeroQ (Cai et al., 2020), DFQ (Choi et al., 2020), and ZAQ (Liu et al., 2021). ZeroQ retrains a quantized model using reconstructed data instead of original data, DFQ is a post-training quantization approach that utilizes a weight equalization scheme to eliminate outliers in both weights and activations, and ZAQ is the pioneering method that employs adversarial learning for data-free quantization. In this comparison, our method consistently demonstrated superior accuracy across all four scenarios.

Table 9: The results of compared methods in Data-free Quantization.

| Dataset | Model | Bit | Float32 | ZeroQ | DFQ | ZAQ | NAYER ($\mathcal{E} = 300$) |
|---|---|---|---|---|---|---|---|
| **CIFAR10** | **MobileNetV2** | W6A6 | 92.39 | 89.9 | 85.43 | 92.15 | **92.23** |
| | **VGG19** | W4A8 | 93.49 | 92.69 | 92.66 | 93.06 | **93.15** |
| **CIFAR100** | **Resnet20** | W5A5 | 69.58 | 65.7 | 59.42 | 67.94 | **68.23** |
| | **Resnet18** | W4A4 | 77.38 | 70.25 | 40.35 | 72.67 | **73.32** |

## B.2    COMPARISON WITH DIFFERENT MEMORY BUFFER SIZE

In this comparison, we evaluate the accuracies of our NAYER (Noisy Label Generator) and MBD-FKD models while varying the memory buffer size. Note that, to ensure a fair and unbiased assessment, we maintain identical generator architectures, including the additional linear layer (noisy layer for NAYER) for both NAYER and MBDFKD. The results demonstrate that: 1) With a bigger memory size, our method can have better performance. 2) Even with only 5k memory size, our method still outperforms the current SOTA DFKD method (90.41% compared to 90.38% of SpaceshipNet).

Table 10: The accuracies of NAYER and MBDFKD with varying the memory buffer size.

| Memory buffer size | 5k | 10k | 20k | 40k | 100k | 200k | Full | SOTA |
|---|---|---|---|---|---|---|---|---|
| MBDFKD | 73.33 | 74.12 | 73.72 | 72.68 | 71.96 | 71.27 | 70.72 | 90.38 |
| NAYER | **90.41** | **90.76** | **90.98** | **91.21** | **91.64** | **91.86** | **91.94** | 90.38 |

## B.3    COMPARISON WITH DIFFERENT GENERATION STEPS

We compare NAYER and FM, both utilizing random noise as input, while adjusting the training steps for their generators. To ensure a level playing field, we use identical generator architectures, including the additional linear layer (noisy layer for NAYER), and train all models for 300 epochs. This approach allows us to assess their performance under consistent conditions and understand how varying the generator training steps impacts their accuracy.

Table 11: The accuracies of our NAYER and FM (which uses random noise as the input) with varying training steps for generators. It's important to note that for a fair comparison, we employ the same generator architectures, including the additional linear layer (noisy layer for NAYER) for FM. Furthermore, all models are trained for 300 epochs

| Generator's training steps | $g = 2$ | $g = 5$ | $g = 10$ | $g = 20$ | $g = 30$ | $g = 40$ | $g = 50$ |
|---|---|---|---|---|---|---|---|
| **FM** | 57.08 | 63.83 | 65.12 | 66.82 | 67.51 | 68.23 | 68.18 |
| **NAYER** | **59.23** | **65.14** | **68.13** | **69.31** | **70.42** | **71.72** | **71.70** |

## B.4    FURTHER COMPARISON WITH DIFFERENT GENERATION STRATEGIES

In this section, we conduct an extensive comparison of various generation strategies. These strategies encompass using only LTE (LTE), random noise (Z), one-hot vectors (OH), the sum of random noise and one-hot vectors (Do et al., 2022) (sum(OH,Z)), the concatenation of Z and one-hot vectors (Luo et al., 2020; Yoo et al., 2019) (cat(OH,Z)), the concatenation of Z and LTE (cat(LTE,Z)), the sum of Z and LTE (cat(LTE,Z)), utilizing our noisy layer for each LTE (LTE+NL(1to1)), a single noisy

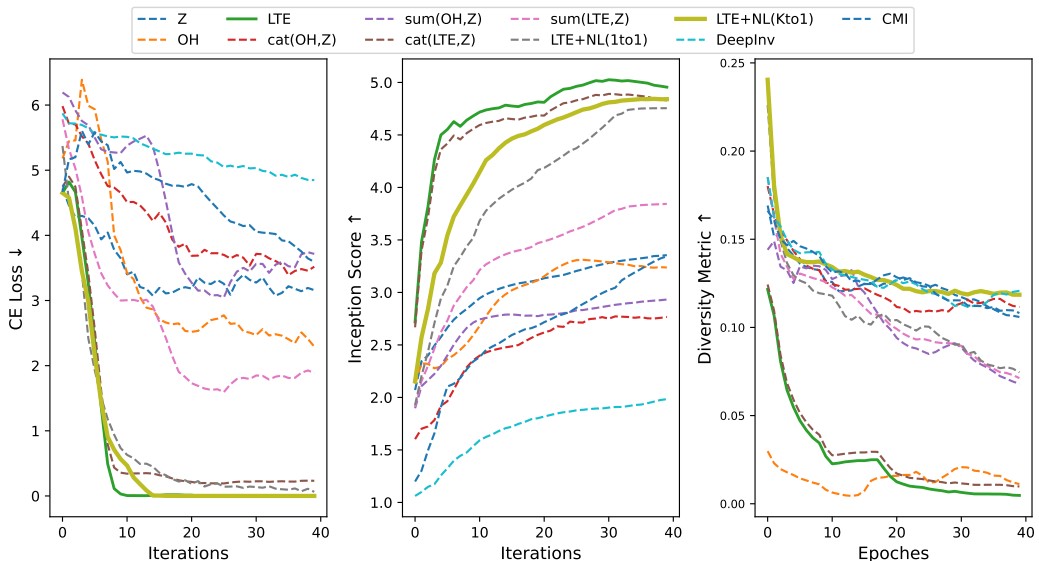

Figure 5: When further comparing various generation strategies based on CE Loss, IS Score, and a diversity metric, it becomes evident that our proposed method, which utilizes the noisy layer for learning LTE, achieves both rapid convergence and high-quality images while maintaining remarkable diversity.

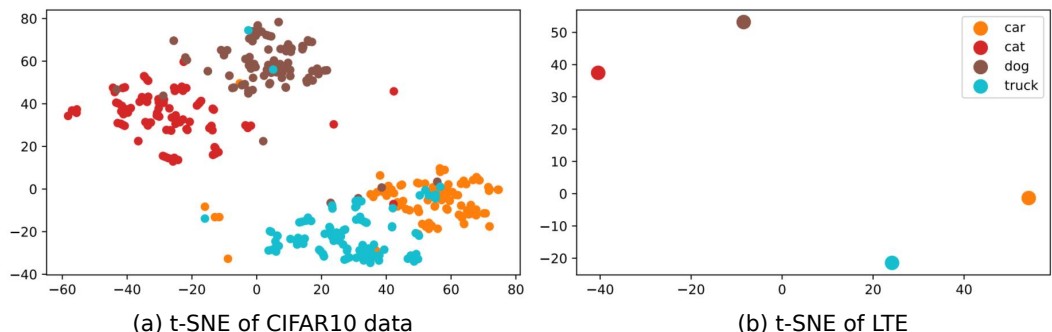

Figure 6: t-SNE Visualization of Label-Text Embedding and Ground-Truth Dataset Distribution for Four Classes: Car, Cat, Dog, and Truck.

layer for all LTE (LTE+NL(1toN)), and the generation methods of DeepInv (Yin et al., 2020) and CMI (Fang et al., 2021). We evaluate these strategies using three metrics: CE Loss to demonstrate convergence speed, IS Score to illustrate image quality, and a diversity metric. It is evident that our proposed method, which leverages the noisy layer for learning LTE, achieves both rapid convergence and high-quality images while maintaining significant diversity.

## B.5    T-SNE VISUALLIZATION OF LTE AND GROUND-TRUTH DATASET DISTRIBUTION

In this section, we aim to illustrate the interclass information captured by LTE (Label-Text Embedding). To achieve this, we provide t-SNE visualizations of the embeddings for labels and ground-truth data distribution pertaining to four distinct classes: Car, Cat, Dog, and Truck. The t-SNE representation of LTE closely aligns with the ground-truth distribution, especially in the proximity between classes like Car and Truck, as well as Cat and Dog, indicating notably smaller distances compared to other class pairings.

## B.6    VISUALIZATION.

The synthetic results achieved by NAYER within just 100 generator training steps on ImageNet by employing the ResNet-50 as teacher model are presented in Fig. 7a-b. For further comparison, we also visualize synthetic images generated by NAYER, FM, CMI, and DeepInv in Fig. 7c-f. All of these samples are generated using 20 steps with a ResNet-34 teacher model in the CIFAR-10 dataset.

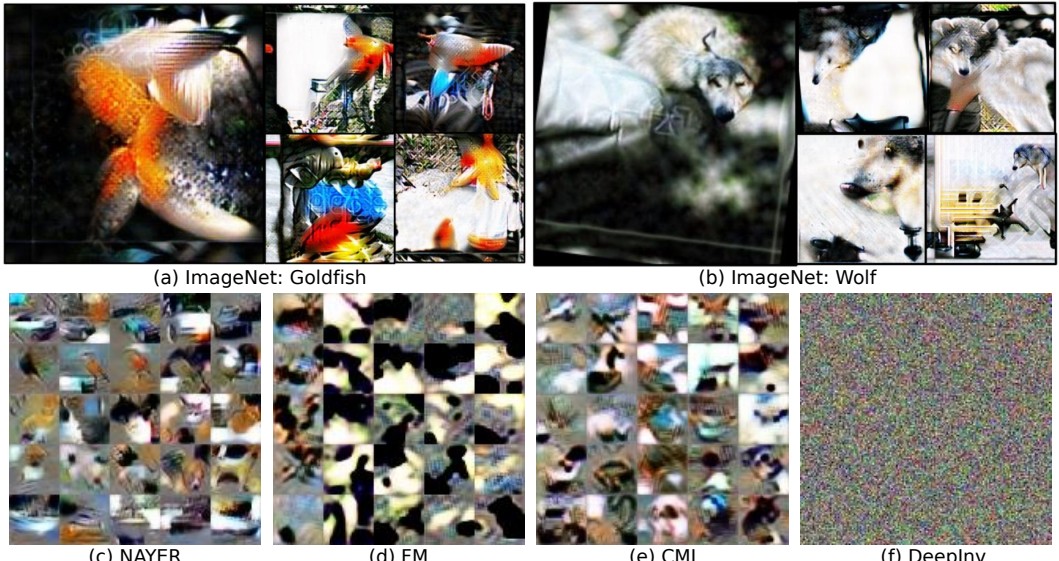

Figure 7: (a, b) Display synthetic data generated by our NAYER for ImageNet in just 100 steps. (c, d, e, f) Showcase synthetic data generated for 5 classes (from top to bottom: Car, Bird, Cat, Dog, Ship) in CIFAR10, using only 20 steps of NAYER, FM, CMI, and DeepInv.

While it remains challenging for human recognition, our method visibly demonstrates superior quality and a more diverse range of images when compared to other methods.

## B.7 FURTURE WORKS

The proposed NAYER does not incorporate the innovative techniques utilized in current SOTA methods, such as feature mixup (Yu et al., 2023), knowledge acquisition and retention (Patel et al., 2023), and momentum updating (Do et al., 2022). This leaves space for potential improvements through the integration of these techniques in the future. Additionally, NAYER can be applied to various data-free methods, including but not limited to data-free quantization or data-free model stealing.

