# OpenReview forum: "Unleash Data Generation for Efficient and Effective Data-free Knowledge Distillation"
_ICLR.cc/2024/Conference — ICLR 2024 Conference Withdrawn Submission_

### Official Review · Reviewer_ovkC · 2023-10-31

**Soundness:** 2 fair
**Presentation:** 2 fair
**Contribution:** 2 fair
**Rating:** 5
**Confidence:** 5

**Summary:**

In this study, the authors address the challenge associated with generating pseudo samples from generators in the adversarial data-free knowledge distillation framework. They introduce the Noisy Layer Generation method (NAYER), an innovative approach that shifts the source of randomness from the input to a designated noisy layer. Instead of traditional inputs, the authors utilize label-text embedding (LTE), which encapsulates significant inter-class distinctions. This strategic incorporation of LTE enables learning high-quality samples faster. Simultaneously, the noisy layer augments sample diversity, ensuring that the model doesn't overly fixate on the label.

**Strengths:**

1. This research stands out as one of the first efforts in DFKD that harnesses a foundational model like CLIP.

2. While current state-of-the-art DFKD methods are often time-intensive and require prolonged training periods for knowledge transfer, the authors convincingly demonstrate in Table 2 that they achieve a marked acceleration. This efficiency is attributed to their use of latent text embeddings, which encode nuanced interclass relationships, thereby enhancing the generation process by exploiting these relationships.


3. Figure 3(d) highlights the generator's undue emphasis on labels within the Adversarial DFKD framework. This observation then drives the authors to inject randomness using noisy layers into the label-text embeddings sourced from CLIP, addressing the identified limitation.


4. Notably, the authors present comparisons and results on expansive datasets like ImageNet, seldom seen in similar works.


5. The 'Extended Results' section, located in the appendix, offers profound insights. It includes rigorous ablation studies, examining various language embeddings and their respective noise-embedding strategies.

**Weaknesses:**

Major Weaknesses:
1. Clarity on Noisy Layers: The proposed method heavily relies on the noisy layers. However, in its present form, the manuscript does not elucidate the mathematical intricacies of the noisy layer, denoted as $Z$. The authors seem to present this layer as an opaque entity. It remains unclear whether the noisy layer employed is analogous to the one proposed by Fortunato et al. [1]. Given the rarity of noisy layer implementations in literature, the authors should elucidate its underpinnings, possibly drawing comparisons to prior work. The primary novelty appears to stem from the introduction of this noisy layer, and without deeper insight into its operations, the paper seems to lack substantial technical contributions.


2. Ambiguities in Section 3.3: The closing remarks of Section 3.3, where the generation of MK synthetic images is broached, seem nebulous. A more comprehensive and systematic exposition of this segment would be beneficial. Furthermore, the authors' decision to employ BatchNorm with embeddings is not clearly justified. It would be insightful to understand the rationale and the potential implications of omitting this step.


3. Memory Overhead Concerns: While the study commendable reduces student training time, there's no discussion on the potential increase in memory overheads that might be attributed to the introduction of noisy layers.


Minor Weaknesses:
1. Citation Misrepresentation in Section 3.1: The initial sentences of the second paragraph of Section 3.1 mistakenly attribute an adversarial mechanism to Nayak et al. (2019). In reality, Nayak and his collaborators proposed a strategy for generating class impressions from pretrained teachers, leveraging these for knowledge distillation.

Reference:

[1] Fortunato, Meire, et al. "Noisy networks for exploration." arXiv preprint arXiv:1706.10295 (2017).

**Questions:**

See Weaknesses

---

> ### Author Response · Authors · 2023-11-20
>
> We appreciate your constructive and thoughtful feedback. Your comments help a lot in improving the quality of the paper. We address your concerns as follows.
>
> **Q1: Clarity on Noisy Layers: The proposed method heavily relies on the noisy layers. However, in its present form, the manuscript does not elucidate the mathematical intricacies of the noisy layer, denoted as . The authors seem to present this layer as an opaque entity. It remains unclear whether the noisy layer employed is analogous to the one proposed by Fortunato et al. [1]. Given the rarity of noisy layer implementations in literature, the authors should elucidate its underpinnings, possibly drawing comparisons to prior work. The primary novelty appears to stem from the introduction of this noisy layer, and without deeper insight into its operations, the paper seems to lack substantial technical contributions.**
>
> **A1:** We have revised almost the proposal method to clarify the motivation and the insight of our method. To summarize, the NL is design as a combination of a BatchNorm layer and a single Linear layer. In that each component have its roles:
> - The simplicity of the single Linear layer is crucial for expediting the generation process. It converges rapidly without requiring an excessive number of steps, yet its size remains sufficiently large to provide an sufficient random source for the generator.
> - Additionally, a BatchNorm module plays a role in increasing the distance between LTEs (from averaging 0.015 to 0.45 using L2 distance), helping the model discriminate these LTEs easier and thereby speeding up the training process. Furthermore, with a different batch of pseudo label, the output of BatchNorm can vary, introducing a slight additional randomness for the generator.
> Unlike the random source used in previous methods. Our NL provide the large amount of randomness, but still can preserves the method’s efficiency due to the straightforward training of LTE.
>
> **Q2: Ambiguities in Section 3.3: The closing remarks of Section 3.3, where the generation of MK synthetic images is broached, seem nebulous. A more comprehensive and systematic exposition of this segment would be beneficial. Furthermore, the authors' decision to employ BatchNorm with embeddings is not clearly justified. It would be insightful to understand the rationale and the potential implications of omitting this step.**
>
> **A2:** We have revised this section for improved readability. In summary, the existing approach involves creating a separate random source for each instance, resulting in similar inputs generating similar samples. In contrast, our proposal involves employing a single noisy layer to learn from all available classes (K-to-1). The fundamental idea is that each class have distinct LTEs. Therefore, by supplying different inputs from K classes, the noisy layer can generate diverse images.
>
> Regarding BatchNorm, for better clarity in illustrating our work, we have revised Sections 3.2 and 3.2. In Section 3.2, we store only the LTE in memory without using BatchNorm. In Section 3.3, BatchNorm is used as part of the Noisy Layer. Additionally, we added a paragraph to highlight the insight behind the use of BatchNorm, as mentioned in A1. To summarize, it helps us to increase the difference between LTE and provide the minor randomness source for generator.
>
> **Q3: Memory Overhead Concerns: While the study commendable reduces student training time, there's no discussion on the potential increase in memory overheads that might be attributed to the introduction of noisy layers.**
>
> **A3:** Thank you so much to point out this concerns. We have conducted the ablation study about the memory size in appendix, Section B.2. From that, the number of memory size will be the option for user, so if they provide the larger disk space, our method provide the better performance. But with only 5k images in memory (around 15Mb), our method still can outperform the current SOTA (90.41\% compared to 90.38\% of SpaceshipNet).
>
> We invite you to reassess the paper, considering the clarifications provided and the results of additional experiments conducted.

---

> > ### Comment · Reviewer_ovkC · 2023-11-22
> >
> > **R1**: Thank you for providing the clarification regarding the function of the noisy layer in this context. It's greatly appreciated.
> >
> > **R2**: In your manuscript, you mention: *"In the existing approach, a separate random source is created for each instance, as similar inputs generate similar samples. In contrast, we propose employing a single noisy layer to learn from all available classes (K-to-1) by inputting..."*. Could you please elaborate on what is referred to as the *existing approach*? Is it a previously established method in the literature that samples using a fixed set of label embeddings? For instance, your approach utilizes CLIP's embeddings, so I'm curious if this is similar to previous methods.
> >
> > **R3**: Could you provide more context regarding the numbers in the table? Specifically, which dataset was utilized to achieve these results, and what teacher-student pairing was implemented, which text language text-embedding was used?, etc.
> >
> > As a suggestion for any potential revisions: It would be immensely helpful if you could use **some form of color highlighting** to distinguish the revised sections in your paper. This would greatly facilitate the re-review process, as it currently can be challenging to discern what has been modified since the last version. This small change could significantly enhance the clarity and effectiveness of your revisions.

---

> ### Author Response · Authors · 2023-11-22
> **Author Response**
>
> Thank you so much for your response.
>
> **Could you please elaborate on what is referred to as the existing approach? Is it a previously established method in the literature that samples using a fixed set of label embeddings? For instance, your approach utilizes CLIP's embeddings, so I'm curious if this is similar to previous methods.**
>
> To the best of our knowledge, we are the first to use label-text embedding for DFKD. The most similar approach is MAD [1], which uses a set of label one-hot vectors as input and still employs a random noise vector as the random source. In that method, they need to provide different random noise vectors (random sources) to generate distinct images. For example, if we need to generate 100 CIFAR100 images, we require 100 different random noise vectors.
>
> In this work, we find that using a single noisy layer to generate all 100 images (corresponding to 100 classes in CIFAR100) reduces the model size and increases generator diversity, ensuring both fast convergence and high-quality sample generation. This enhancement can be attributed to the use of multiple gradient sources from diverse classes, which further enriches the diversity of the noisy layer's output.
>
> **Could you provide more context regarding the numbers in the table?**
>
> This model is trained on the CIFAR100 dataset using the WideResnet40-2/WideResnet16-2 pair, and all compared methods are trained for 100 epochs. We have also revised the table caption to reflect this information.
>
> **Highlight color.**
>
> Thank you for pointing out this issue. We have submitted the highlight color version.
>
> Finally, we hope that you may consider reevaluating the paper, considering the clarifications provided and the results of additional experiments conducted.
>
> [1] Do, Kien, et al. "Momentum Adversarial Distillation: Handling Large Distribution Shifts in Data-Free Knowledge Distillation." A paper in Neural Information Processing Systems 35 (2022): 10055-10067.

---

### Official Review · Reviewer_m3cF · 2023-11-02

**Soundness:** 3 good
**Presentation:** 3 good
**Contribution:** 2 fair
**Rating:** 5
**Confidence:** 4

**Summary:**

This paper explores a more effective data-free knowledge distillation. The authors believe that the previous DFKD-based generation generated samples from random noises, so no very effective information was extracted. Therefore, in the paper, the authors introduce a novel Noisy Layer Generation method (NAYER) that relocates the randomness source from the input to a noisy layer and utilizes the meaningful label-text embedding (LTE) as the input. Based on this method, this work It can achieve high efficiency while ensuring the diversity of data generation. Extensive experiments illustrate the effectiveness of the proposed method.

**Strengths:**

1. The authors focus on the relationship between the diversity and efficiency of sample generation, which is important for DFKD.

2. The code is released.

**Weaknesses:**

1. CLIP is introduced in this paper, and the training of CLIP requires a large amount of additional data. Other comparison methods do not seem to introduce additional data and only use teachers, which seems to be an unfair comparison. More explanation is needed here.

2. Lack of comparison with sampling-based methods [1][2]. More importantly, there is also design about noise in DFND[1]. Although not exactly the same, it should warrant comparison and discussion.

3. There is a lack of comparison with [3] in terms of generation efficiency. In addition, the distillation performance is not as good as [1][2].

[1] Learning Student Networks in the Wild, CVPR 2021

[2] Sampling to Distill: Knowledge Transfer from Open-World Data, arxiv 2023

[3] Up to 100$\times$ Faster Data-free Knowledge Distillation, AAAI 2022

**Questions:**

See Weaknesses.

---

> ### Author Response · Authors · 2023-11-20
> **Author Response**
>
> We thank you for your reviews and address your concerns as follows.
>
> **Q1: CLIP is introduced in this paper, and the training of CLIP requires a large amount of additional data. Other comparison methods do not seem to introduce additional data and only use teachers, which seems to be an unfair comparison. More explanation is needed here.**
>
> **A1:** Our method does not utilize CLIP in the model training process. We query the LTE only once before training the model, and this LTE is then stored in memory (a small tensor with dimensions of 100x512). During the model training, we exclusively use the LTE stored in memory for data-free generation and do not involve CLIP in this phase. The only requirement is the label description of the dataset, which does not compromise privacy and adheres to the 'data-free' setting. Additionally, we do not use any additional data to adhere to data-free generation principles.
>
> **Q2: Lack of comparison with sampling-based methods [1][2]. More importantly, there is also design about noise in DFND[1]. Although not exactly the same, it should warrant comparison and discussion. In addition, the distillation performance is not as good as [1][2].**
>
> **A2:** Thank you for pointing out the need for references. In response to your comments, we have included a discussion about our method in the context of related work. In summary, [1] and [2] leverage additional data to enhance distillation performance. However, the practical application of these approaches is limited in scenarios involving sensitive or private data, where acquiring suitable unlabeled sources can be challenging.
> In comparison, our method achieves significantly better results than [1] (95.21% and 77.54% compared to 94.02% and 76.32%, respectively, for [1]). Regarding [2], a fair comparison is challenging since they incorporate an additional 600k images from the internet to enhance distillation, while our method do not use them. We attempted to implement our method without these additional data, but unfortunately, their code is not available.
> [1] 'Learning Student Networks in the Wild,' CVPR 2021
> [2] 'Sampling to Distill: Knowledge Transfer from Open-World Data,' arXiv 2023"
>
> **Q3: There is a lack of comparison with [3] in terms of generation efficiency.**
>
> **A3:** We conducted experiments to compare our method with [3] (FM in our paper). Firstly, in Table 2, with similar training times, our NAYER consistently achieves higher accuracy compared to [3] (approximately 3%). Secondly, in Table 11, we also compare our method with [3] at different generator training steps, ranging from 2 to 50. The results demonstrate that our method outperforms [3] in all comparison cases.
>
> We would appreciate it if you could reconsider the paper, taking into account the provided clarifications and the results of additional experiments we have undertaken.

---

> > ### Comment · Reviewer_m3cF · 2023-11-22
> >
> > We still have reservations about the introduction of CLIP because it is trained on large-scale datasets where additional data exists. If this holds, then the comparison of sampling methods [1][2] is also reasonable.

---

> ### Author Response · Authors · 2023-11-22
> **Author Response**
>
> Dear reviewer,
>
> Thank you for your response. Using the foundation model such as CLIP, ViT, Stable Diffusion Model to fine-tune to new tasks is very popular and widely-accepted in our field. For instance, prompt-based tuning works rely on the state-of-the-art transformer-based models such as BERT, ViT, and more to add additional prompts to adapt new tasks. As such, you can observe so many works in NLP and computer vision of prompt-based tuning.
>
> In our work, we use the CLIP only one time to work out the text embedding of the labels. Different from many prompt-based tuning models which heavily rely on the foundation model, our work relies very lightly on CLIP. Additionally, we demonstrate that if we only condition on CLIP embeddings to generate synthetic images, the diversity of generated images is reduced. Hence, we propose the noisy layers. Therefore, our work is not a simple use of CLIP embeddings.
>
> Moreover, we would like to clarify that our method utilizes the power of the text embedding technique, considering CLIP's text encoder as one of the options. Our approach also gets the SOTA results with other language models, such as SBERT and Doc2Vec, where additional images are not available.
>
> Thank you again, and we would appreciate it if you could reconsider the paper.

---

### Official Review · Reviewer_Dv9s · 2023-11-03

**Soundness:** 3 good
**Presentation:** 3 good
**Contribution:** 3 good
**Rating:** 6
**Confidence:** 4

**Summary:**

This paper proposes a new method for data free KD method, Noisy Layer Generation (NAYER), which relocates the randomness source from the input to a noisy layer and utilizes the meaningful label-text embedding (LTE) as the input. LTE, generated by a pretrained text encoder, contains meaningful inter-class information, that enables the generation of high-quality samples with only a few training steps. LTE layer is initialized in each iteration for the diversity of generated images. Experiments suggest the proposed method outperforms other counterparts while being training faster too.

**Strengths:**

1. The idea of using pretrained text encoder to generate label-text embeddings as input for distillation is interesting and sounds novel to me.

2. To achieve diverse generated samples (which is a key problem in DFKD), they propose a noisy layer between the input and the generator. The noisy layer is initialized in each iteration, which effectively introduces more diversity for the synthetic samples.

3. The proposed method not only outperforms other DFKD approaches in terms of student performance but also is much faster in training.

**Weaknesses:**

1. Since this paper utilizes the pretrained text encoder in CLIP model, I think a similar idea for DFKD is to use pretrained text-to-image generation models, such as stable diffusion to generate pseudo data for distillation. It is advisable to add a set of comparison experiments to show the performance difference.

2. What is the effect of reinitializing the noisy layer in each iteration? What if it is not reinitialized? This key ablation study is missing now.

3. The presentation has some small issues to fix:

3.1 The text in Fig. 4 is too small, hard to make out.

3.2 Eq. (6) and (7) should have some punctuation. Make them in a sentence, not orphaned.

3.3 Missing period in the caption of Fig. 5.

**Questions:**

How many images are stored in the memory module M in each epoch? Does this affect the performance significantly (any ablation study about it)?

---

> ### Author Response · Authors · 2023-11-20
> **Author Response**
>
> We thank you for your reviews and address your concerns as follows.
>
> **Q1: Since this paper utilizes the pretrained text encoder in CLIP model, I think a similar idea for DFKD is to use pretrained text-to-image generation models, such as stable diffusion to generate pseudo data for distillation. It is advisable to add a set of comparison experiments to show the performance difference.**
>
> **A1:** Our method significantly differs from using pretrained text-to-images generation to generate pseudo-data. In our approach, we only query the LTE from the language model (LM) once. This LTE is then stored in memory for subsequent processing, and we do not utilize the language model in the training process. Therefore, comparing our method with these approaches is unfair, as we do not rely on any additional data. Furthermore, some methods using pretrained text-to-image may face challenges when dealing with cases involving privacy concerns, such as in medical scenarios where images are not typically publicly available on the internet or in the training set of these text-to-image generation models.
>
> **Q2: What is the effect of reinitializing the noisy layer in each iteration? What if it is not reinitialized? This key ablation study is missing now.**
>
> **A2:** In our method, reinitializing the noisy layer serves as the primary random source for generating different sets of images. Consequently, without this reinitialization, our method would produce similar images across iterations. To further illustrate this, we conducted an additional ablation study, and the results presented in Table 4 show that without reinitiation, the Noisy Layer (NAYER) provides nearly identical results to using LTE alone. This underscores the effectiveness of reinitiation strategies.
>
> **Q3: The presentation has some small issues to fix:**
>
> **A3:** Thank you so much; we appreciate your constructive and thoughtful feedback. We have implemented all of your suggestions.
>
> We kindly request that you reconsider the paper in light of these clarifications and the additional experiments we have conducted.

---

> > ### Comment · Reviewer_Dv9s · 2023-11-22
> > **Thanks for the feedback.**
> >
> > **Q1:**
> >
> > I guess I mentioned using text-to-image generation for synthesizing pseudo data because it is an obvious way to address the DFKD problem today. I agree that the presented method "*significantly differs from using pretrained text-to-images generation to generate pseudo-data*". But honestly, for someone who really wants to make DFKD *practical*, what I mentioned is at least a very triable approach. The authors seem not to agree with this, unfortunately.
> >
> > “*Therefore, comparing our method with these approaches is unfair, as we do not rely on any additional data.*" -- Again, it shows a difference between "we want to resolve the problem" and "we just want to write a paper.". Not relying on additional data is good, but if you do have access to additional (synthetic) data, why not use it? Esp. using the additional data is probably the most promising way to achieve a good performance these days. Regarding this claim "*we do not rely on any additional data*", if we consider the LTE idea as implicitly taking advantage of the foundation model *that is trained on a large corpus of data*, this method also relies on additional data *implicitly*.
> >
> > "*Furthermore, some methods using pretrained text-to-image may face challenges when dealing with cases involving privacy concerns, such as in medical scenarios where images are not typically publicly available on the internet or in the training set of these text-to-image generation models.*" -- I see a lot of DFKD papers use this privacy issue as motivation. Indeed, it is true, generally speaking. However, this work, like many others, does not have empirical results on these privacy datasets to support this. I would suggest not making this claim unless you have solid results.
> >
> > **Q2:**
> >
> > Do you mean Tab. 3? Table 4 shows the accuracies of our NAYER with three different text encoders.

---

> ### Author Response · Authors · 2023-11-22
> **Author Response**
>
> Thank you so much for your response.
>
> **Q1:** We believe that employing text-to-image generation approaches is a viable and highly promising solution for the DFKD problem. However, due to limitations in resources and time, we were unable to conduct a more extensive comparison with these methods at this time. Therefore, we consider them as potential approaches for future research.
>
> **Q2:** Thank you for pointing this out, this is correctly in Table 3.
>
> Finally, we hope that you may consider reevaluating the paper.

---

### Official Review · Reviewer_axyR · 2023-11-10

**Soundness:** 3 good
**Presentation:** 2 fair
**Contribution:** 2 fair
**Rating:** 3
**Confidence:** 4

**Summary:**

Data-Free Knowledge Distillation (DFKD) has made significant strides in recent years, with its core principle of transferring knowledge from a teacher neural network to a student neural network without requiring access to the original data. However, existing approaches face a major challenge when attempting to generate samples from random noise inputs, which lack meaningful information. As a result, these models struggle to effectively map this noise to the ground-truth sample distribution, leading to low-quality data and substantial time requirements for training the generator.

To address this issue, this paper proposes al Noisy Layer Generation method (NAYER) that relocates the randomness source from the input to a noisy layer and utilizes the meaningful label-text embedding (LTE) as the input. The significance of LTE lies in its ability to contain substantial meaningful inter-class information, enabling the generation of high-quality samples with only a few training steps. Simultaneously, the noisy layer plays a key role in addressing the issue of diversity in sample generation by preventing the model from overemphasizing the constrained label information. By reinitializing the noisy layer in each iteration, this work aims to facilitate the generation of diverse samples while still retaining the method’s efficiency.

**Strengths:**

- The paper introduces a DFKD method called Noisy LAYER Generation (NAYER) that relocates the source of randomness from the input to the noisy layer and utilizes label-text embedding (LTE) as the input. Using LTE as input allows for proficient generation of high-quality samples that closely mimic the distributions of their respective classes with only a few training steps.
- Extensive evaluation is presented on standard benchmark datasets like CIFAR10, CIFAR100, TinyImageNet, and ImageNet. It achieves superior performance against the prior arts.

**Weaknesses:**

1) In the introduction section, paragraphs 4 and 5 seem disconnected from the rest of the introduction and disrupt the flow of reading. There are seveal references to different figures and experimental results that are presented in other pages and also in supplementary. This disturbs the flow of reading. The authors should rewrite these paragraphs in a way that simplifies the key ideas, potentially using examples to make them more accessible to a broader audience. It's important for the introduction to provide a smooth and coherent overview of the paper's content to engage a wider audience.

2) The paper proposes the use of label-text embedding (LTE), which may have a disadvantage compared to other methods that do not rely on such embedding. For example, in cases like classifying chemical compounds, where label-text embedding, like CLIP, may not be applicable. Other data-free knowledge distillation methods do not rely on such joint image-text embedding knowledge thus they world perform reasonably well for a wide variety of classification modalities (such as audio, chemical-compund, etc.). Highlighting the limitations of the proposed approach is essential for a balanced evaluation of its potential use cases.

3) The paper uses label-text embedding (LTE) obtained from a pre-trained model CLIP that is trained on image-text pairs from the internet. Drawing a comparison to CLIP, which also has knowledge of common objects and corresponding text, suggests that the proposed method is not entirely data-free. This raises doubts about whether it can be truly considered "data-free" distillation.

4) Limited novely: The proposed approach builds upon CLIP embedding for data-free knowledge distillation but offers limited innovation. Section 3.1 appears to reiterate the existing Data-free knowledge distillation framework without introducing fresh perspectives or insights beyond what has been discussed in prior works. Further, the use of label-text embedding followed by a randomly initialized layer makes the generator resemble a conditional GAN. There are various alternative ways to achieve the desired setup, such as: a) employing a conditional GAN where a noise vector-based embedding and an LTE-based embedding are concatenated to form the initial part of the generator network, b) using an equation like e + beta*(z~ N(0, I)) to model intra-class diversity and span the embedding space between classwise LTE embeddings, or c) exploring some form of linear combination of LTE embedding with specialized weight sampling. The inclusion of the proposed Noise Layer appears unnecessary and lacks clear justification.

**Questions:**

Please see the Weaknesses section.

---

> ### Author Response · Authors · 2023-11-20
> **Author Response**
>
> We thank you for your reviews and address your concerns as follows.
>
> **Q1: In the introduction section, paragraphs 4 and 5.**
>
> **A1:** I greatly appreciate your constructive and thoughtful feedback. Following your suggestion, we have enhanced the flow and coherence of paragraphs 4 and 5, incorporating relevant examples for a more seamless presentation.
>
> **Q2:  The limitations of the proposed approach is essential for a balanced evaluation of its potential use cases.**
>
> **A2:** Thank you for highlighting the potential issue. In our paper, we emphasize two primary advantages of utilizing LTE. Firstly, LTE serves as a dense vector that encompasses more information, facilitating an easier learning process for the model. Secondly, LTE possesses the capacity to illustrate relationships between classes. In instances where a class lacks a meaningful label, we can still employ the class index to generate a label text, such as P3:"a class of class_index," enabling the discovery of LTE. Although this approach may not fully capture the relationships between classes, it still contains richer information compared to a one-hot vector. To substantiate this claim, we conducted an ablation study on various prompt engineering techniques in Section 4.4. The results demonstrate that when using only the label index instead of the label name, the performance of P3 remains significantly superior to the best baseline. Specifically, this template P3 achieves 93.72% accuracy compared to SpaceshipNet (the SOTA) with 93.25%, and 71.17% compared to 69.95%, respectively. This underscores the viability of employing the label index, particularly in datasets with less meaningful labels, further validating the effectiveness of our methods in real-world applications.
>
> **Q3: The doubts about whether it can be truly considered "data-free" distillation.**
>
> **A3:** The foundation of our work relied on the dense and richer information LTE produced any pretrained language model (LM). To demonstrate this claim, we conducted an ablation study with SBERT, Doc2Vec and CLIP. This originally located in the appendix, which has now been integrated into the main paper as Section 4.4 for enhanced accessibility. The results reveal that CLIP exhibits a only minor improvement (approximately 0.09%) compared to other LMs, suggesting that our method is compatible with various LM. It is important to note that we query LTE from the LM only once and store it in memory for model training; thereafter, the LM is no longer used in the training process. Additionally, our approach adheres to a "data-free" setting, as we solely rely on the label description of the task and refrain from using any training data.
>
> **Q4: Limited novely.**
>
> **A4:** Thank you for your questions. While we respect your viewpoint, it does not accurately reflect the novel contributions that exists in the paper. Our work introduces several novel methodological contributions.
>
> (1) To my best knowledge we are the first to introduce the LTE for data-free knowledge distillation.
>
> (2) We also the first to introduce the layer-level random source for this setting to address the limitation of two common random source including the concatenation and the sum (as your mentioned):
>
> - Concatenation raises the risk of overemphasizing label-related information, as evidenced by significantly larger weight magnitudes for learning LTE compared to random noise, which can be seen as the significance of these weights
>
> - Using the sum faces challenges: a low $\beta$ results in an insufficient random source for diverse sampling, and a high $\beta$ may overshadow LTE features, leading to a reliance on random noise. This challenge is also observed in some existing methods, where the application of the sum of noise and label information provides minimal improvement compared to an unconditional generator.
>
> In light of this, it demonstrates the benefits of the Noisy Layer, which can successfully mitigate the risk of a negative bias towards LTE and provide a larger random parameter to enhance the diversity of the synthesized images. We also conducted the ablation study to compare them at Section 4.3. The results demonstrate that our Noisy Layer further better the concatenation and the sum (at least 3%).
>
> Furthermore, our experiments demonstrate that our work not only has state-of-the-art (SOTA) performance on DFKD but also can accelerate training time from 5x to 15x, further highlighting the contributions of our work.
>
> We hope that you may consider reevaluating the paper in light these clarifications and additional experiments that we have performed.

---

### Author Response · Authors · 2023-11-20
**General Response**

We express our gratitude to all the reviewers for their valuable feedback. Drawing inspiration from their comments, we have made modifications to the text and conducted several additional experiments. The outcomes of these adjustments are now incorporated into the updated version of the paper.

(1) We have enhanced the smoothness and coherence of paragraphs 4 and 5.

(2) We have reorganized and rewritten a significant portion of the proposal method section, incorporating changes to the architecture image and pseudo-code to better illustrate our ideas. The main highlights include:

- Our method queries label-text embedding (LTE) from the language model (LM) only once and stores it in memory for all subsequent training processes. The LM is not used or fine-tuned in model training.
- Our method demonstrates compatibility with any LM, including SBERT, Doc2Vec, and CLIP. The benefits from foundation models, such as CLIP, improve our accuracy, albeit only to a minor extent.
- When using only the class index instead of class description, our method still outperforms the state-of-the-art (SOTA) in data-free knowledge distillation.

(3) We conducted additional ablation studies to illustrate our claims that our work functions effectively with any pretrained language model and that the improvement in the foundation model of CLIP in our work is only minor.

(4)  Further ablation studies were conducted to provide a more in-depth analysis of our work with different prompt engineering templates.

(5)  We have reorganized some important ablation studies from the appendix to the main paper for enhanced accessibility.

We hope that our response and updates can be carefully considered.